# Reasoning Models Hallucinate More: Factuality-Aware Reinforcement Learning for Large Reasoning Models

**Junyi Li**      **Hwee Tou Ng**
Department of Computer Science, National University of Singapore
cheneyjunyi@gmail.com, nght@comp.nus.edu.sg

## Abstract

Large language models (LLMs) have significantly advanced in reasoning tasks through reinforcement learning (RL) optimization, achieving impressive capabilities across various challenging benchmarks. However, our empirical analysis reveals a critical drawback: reasoning-oriented RL fine-tuning significantly increases the prevalence of hallucinations. We theoretically analyze the RL training dynamics, identifying high-variance gradient, entropy-induced randomness, and susceptibility to spurious local optima as key factors leading to hallucinations. To address this drawback, we propose Factuality-aware Step-wise Policy Optimization (**FSPO**), an innovative RL fine-tuning algorithm incorporating explicit factuality verification at each reasoning step. FSPO leverages automated verification against given evidence to dynamically adjust token-level advantage values, incentivizing factual correctness throughout the reasoning process. Experiments across mathematical reasoning and hallucination benchmarks using Qwen2.5 and Llama models demonstrate that FSPO effectively reduces hallucinations while enhancing reasoning accuracy, substantially improving both reliability and performance.

⌂ https://github.com/nusnlp/FSPO

## 1 Introduction

Large language models (LLMs) have recently achieved remarkable breakthroughs in natural language generation and complex reasoning tasks [9, 15]. This progress is fueled by new training paradigms that imbue LLMs with reasoning capabilities, including fine-tuning on long chain-of-thought (CoT) solutions [43] and reinforcement learning (RL) based optimization [9]. By leveraging these approaches, state-of-the-art models can decompose complex problems into reasoning steps and iteratively refine their answers, yielding impressive results on challenging benchmarks in mathematics [10], multi-hop question answering [41], coding [2], and decision-making [32].

Along with improved reasoning ability, a critical question has emerged: *To what extent do reasoning models maintain factuality after RL fine-tuning?* In this work, our empirical observations indicate that when LLMs are trained with outcome-driven RL, they tend to produce more factually incorrect or fabricated statements (i.e., hallucinations [13, 46]) during their reasoning process. We find that these hallucinations usually stem from incorrect intermediate reasoning, i.e., the model may generate unsupported or false claims in the chain-of-thought, even if its final answer is occasionally correct. Indeed, RL-aligned models can be highly fluent and confident in their step-by-step explanations while subtly introducing factual errors [40]. This issue undermines the reliability of RL-trained reasoning models, as a flawed or untruthful answer is problematic for trust and interpretability.

To understand why outcome-based RL fine-tuning exacerbates hallucinations, we conduct a theoretical analysis of the RL training dynamics for reasoning tasks. Our analysis identifies three primary factors.

39th Conference on Neural Information Processing Systems (NeurIPS 2025).

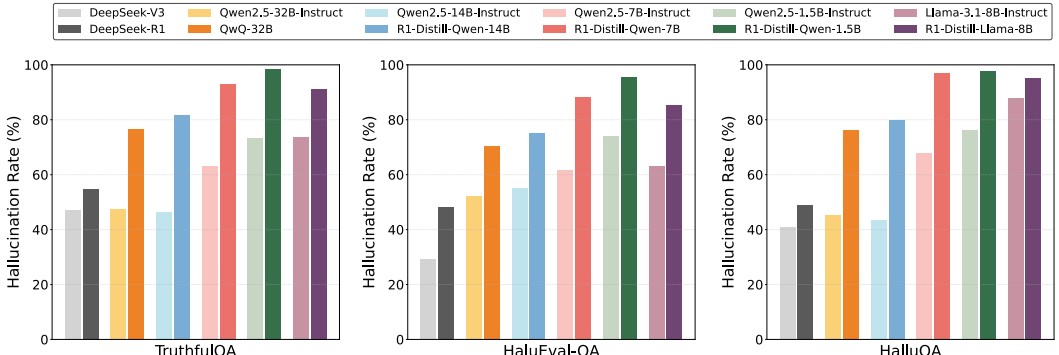

Figure 1: Hallucination rate (%) for six groups of twelve LLMs across three benchmarks.

First, optimizing solely for correct final answers leads to extremely high variance in the policy gradient when such answers are rare, yielding unstable training updates. Second, the need to discover rewarding outputs forces the policy to maintain a high level of entropy in its predictions, heightening the risk of hallucinations. Third, standard RL objectives admit spurious local optima whereby the model can converge to a confidently held wrong answer that yields zero reward. These three factors, i.e., high-variance gradient, entropy-induced randomness, and spurious local optima, together explain why a purely outcome-based RL approach can cause a reasoning model to hallucinate and struggle to learn faithful reasoning patterns.

To mitigate these issues, we propose **F**actuality-aware **S**tep-wise **P**olicy **O**ptimization (**FSPO**), a novel RL fine-tuning algorithm that explicitly incorporates factuality feedback at each reasoning step. The key idea is to adjust the advantage values of individual tokens using a step-wise factuality reward signal, encouraging LLMs to generate more factual content. Specially, we employ an automated verifier to check whether each generated reasoning sentence is entailed by the given evidence, yielding step-wise factuality scores. These scores are then integrated into the overall reward signal to adjust the advantages of tokens during optimization, rewarding factually correct tokens while penalizing incorrect tokens. This factuality-aware reward shaping provides denser and more informative feedback to the policy, addressing the sparse-signal problem and reducing training instability. FSPO steers the policy toward solutions that are not only correct but also generated via faithful, verifiable chains of reasoning, thereby directly mitigating hallucinations in the RL training process.

We conduct extensive experiments to evaluate the effectiveness of FSPO on both mathematical reasoning and hallucination benchmarks using Qwen2.5-7B-Base/Instruct and Llama-3.1-8B-Instruct. The results show that our method achieves significantly improved mathematical reasoning performance and reduced hallucinations. Further detailed analyses indicate that our method can enhance the factuality of reasoning steps without compromising the quality of generated content.

## 2 Preliminary Experiments

To explore the effect of RL on hallucinations, we conduct preliminary experiments based on several LLMs and hallucination benchmarks.

**Models and Datasets.** For comparative analysis, we select several groups of LLMs without and with large-scale RL training: (1) *DeepSeek-V3* [21] and *DeepSeek-R1* [9]; (2) *Qwen2.5-32B-Instruct* [39] and *QwQ-32B* [28]; (3) *Qwen2.5-14B/7B/1.5B-Instruct* [39] and *R1-Distill-Qwen-14B/7B/1.5B* [9]; (4) *Llama3.1-8B-Instruct* [8] and *R1-Distill-Llama-8B [9]*. These distilled models are fine-tuned on the synthesized long-CoT data from DeepSeek-R1, thus exhibiting similar thinking modes to RL-tuned models. For the hallucination datasets, we select three benchmarks, namely *TruthfulQA* [20], *HaluEval* [19], and *HalluQA* [4]. In TruthfulQA, we adopt the generation task, where the model is prompted to generate an answer to a question, and then utilize a fine-tuned GPT-3 to predict the truthfulness of the answer. The hallucination rate is equal to the ratio of untruthful answers. HalluQA follows the same method as TruthfulQA, except that it employs GPT-4 as the evaluator. For HaluEval, we adopt the QA subset, where the LLM is provided with a question and two candidate answers and

it needs to assess which answer is factually correct. The hallucination rate refers to the ratio of false judgments made by the model.

**Results and Analysis.** The results are shown in Figure 1. We can clearly observe that after large-scale training with RL or long CoT data, models demonstrate dramatically higher degree of hallucinations across the three benchmarks. To further investigate the underlying causes of this phenomenon, we randomly select 200 samples from HaluEval-QA that DeepSeek-V3 answers correctly while DeepSeek-R1 does not. We manually examine these cases and classify the source of errors (refer to Appendix E for more details). In Figure 2, we can see that most errors stem from incorrect reasoning steps. We hypothesize that this may be attributed to the outcome reward modeling, where rewards and penalties are assigned solely based on the correctness of the final answer, while ignoring signals from intermediate reasoning steps. In the following sections, we will provide a theoretical analysis of how outcome-based reward modeling in RL could lead to hallucinations, and then propose the factuality-aware step-wise RL algorithm to mitigate this issue.

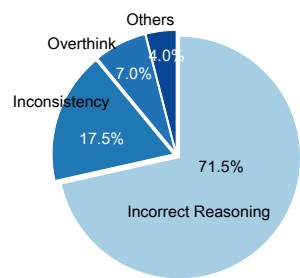

Figure 2: Distribution of hallucination causes over 200 samples from HaluEval-QA.

## 3 Background

**Reinforcement Learning.** In reinforcement learning, language model generation can be formulated as a token-level Markov decision process, denoted by $\mathcal{M} = \{\mathcal{S}, \mathcal{A}, \mathcal{R}, \pi_\theta\}$. At the $t$-th generation step, the state $s_t \in \mathcal{S}$ is the concatenation of the input prompt $x$ and the generated sequence so far $y_{<t} = [o_1, ..., o_{t-1}]$. The policy $\pi_\theta(\cdot|s_t)$ (parameterized by $\theta$) selects the next action $o_t \in \mathcal{A}$ by generating a token from the vocabulary $\mathcal{A}$, leading to a deterministic transition to the next state $s_{t+1} = [s_t, o_t]$. Let $Y^*$ denote the set of correct answers for a given input prompt $x$. Existing research mostly adopts a binary reward $\mathcal{R}$ for each output $y$, which is sparse and discontinuous [32, 18]. For simplicity, we consider a prevalent format in this paper: $\mathcal{R}(y) = 1$ when $y \in Y^*$, or $\mathcal{R}(y) = 0$ when $y \notin Y^*$. The expected return (reward) under the policy $\pi_\theta$ is:

$$\mathcal{J}(\theta) = \mathbb{E}_{y \sim \pi_\theta(\cdot|x)}[\mathcal{R}(y)] = \sum_y \mathcal{R}(y)\pi_\theta(y|x). \tag{1}$$

With the binary reward, the expected return is essentially the probability of producing a correct output under the current policy $\pi_\theta$. In the context of RL, we can update the policy using policy gradient algorithms. According to the policy gradient, the gradient of this objective can be written as:

$$\nabla_\theta \mathcal{J}(\theta) = \mathbb{E}_{y \sim \pi_\theta(\cdot|x)}[\mathcal{R}(y) \nabla_\theta \log \pi_\theta(y|x)]. \tag{2}$$

The policy gradient methods maximize the expected total reward by repeatedly estimating the gradient. Several policy gradient algorithms have been proposed such as Proximal Policy Optimization [29].

**Group Relative Policy Optimization.** In DeepSeek-Math [30], researchers proposed a new policy gradient method, Group Relative Policy Optimization (GRPO), which improves the effectiveness and efficiency of RL training based on Proximal Policy Optimization (PPO) [29]. The PPO algorithm maximizes the following surrogate objective:

$$\mathcal{J}_{\text{PPO}}(\theta) = \mathbb{E}_{y \sim \pi_\theta(\cdot|x)} \sum_{t=1}^{|y|} \left\{ \min\left[ \frac{\pi_\theta(o_t|x, y_{<t})}{\pi_{\theta_{\text{old}}}(o_t|x, y_{<t})} A_t, \text{clip}\left( \frac{\pi_\theta(o_t|x, y_{<t})}{\pi_{\theta_{\text{old}}}(o_t|x, y_{<t})}, 1 - \epsilon, 1 + \epsilon \right) A_t \right] \right\}, \tag{3}$$

where $\pi_{\theta_{\text{old}}}$ is the policy before the update, $\epsilon$ is the clip hyperparameter, and $A_t$ is the advantage of the $t$-th token. In PPO, a value model is learned to estimate the advantage based on the reward $\mathcal{R}$, which is computationally expensive. Thus, GRPO uses the average reward of multiple sampled outputs as baseline in advantage estimation. Specifically, for each input $x$, GRPO samples a group of outputs $\{y_1, ..., y_G\}$ and computes their rewards $\{\mathcal{R}_1, ..., \mathcal{R}_G\}$, then sets the advantage of all tokens from $y_i$ as $A_{i,t} = (\mathcal{R}_i - \text{mean}(\{\mathcal{R}_1, ..., \mathcal{R}_G\}))/\text{std}(\{\mathcal{R}_1, ..., \mathcal{R}_G\})$. The GRPO objective can be written as:

$$\mathcal{J}_{\text{GRPO}}(\theta) = \mathbb{E}_{y \sim \pi(\cdot|x)}$$

$$\frac{1}{G} \sum_{i=1}^{G} \frac{1}{|y_i|} \sum_{t=1}^{|y_i|} \left\{ \min\left[ \frac{\pi_\theta(o_{i,t}|x, y_{i,<t})}{\pi_{\theta_{\text{old}}}(o_{i,t}|x, y_{i,<t})} A_{i,t}, \text{clip}\left( \frac{\pi_\theta(o_{i,t}|x, y_{i,<t})}{\pi_{\theta_{\text{old}}}(o_{i,t}|x, y_{i,<t})}, 1 - \epsilon, 1 + \epsilon \right) A_{i,t} \right] \right\}. \tag{4}$$

Compared to PPO, GRPO foregoes the value model to estimate advantages that is typically the same size as the policy model, relieving the memory and computational burden.

## 4 Theoretical Analysis

In this section, we present a theoretical analysis that characterizes how the simplified and prevalent binary reward format (i.e., 1/0) affects the learning dynamics and induces hallucinations. The analysis can extend to any arbitrary bounded real rewards.

**Theorem 4.1** (Variance of Policy Gradient in Binary Reward RL). *Under a binary reward function, the variance of the policy gradient estimator is directly proportional to the probability of generating correct answers. Let $p$ denote the probability of producing a correct answer, so the variance of the policy gradient estimator is on the order of $p(1-p)\|\nabla_\theta \log \pi_\theta(y^*|x)\|^2$ (where $y^*$ is a correct answer). For small $p$, this variance is extremely high, hindering stable learning.*

*Proof.* The REINFORCE gradient for a given input is $g = R(y)\,\nabla_\theta \log \pi_\theta(y|x)$. This is a random vector depending on the sampled output $y$. There is a $p$ chance that the sampled output is correct (yielding $R = 1$) and a $1 - p$ chance that it is wrong (yielding $R = 0$). Hence, the gradient $g$ is zero with probability $1 - p$, and equals $\nabla_\theta \log \pi_\theta(y^*|x)$ with probability $p$. The expected gradient is $\mathbb{E}[g] = p\,\nabla_\theta \log \pi_\theta(y^*|x)$. The second moment is $\mathbb{E}[\|g\|^2] = p\,\|\nabla_\theta \log \pi_\theta(y^*|x)\|^2$. Therefore, the covariance (and in particular the trace as a measure of variance magnitude) is on the order of

$$p\,\|\nabla_\theta \log \pi_\theta(y^*|x)\|^2 - \|p\,\nabla_\theta \log \pi_\theta(y^*|x)\|^2 = p(1-p)\,\|\nabla_\theta \log \pi_\theta(y^*|x)\|^2. \tag{5}$$

For very small $p$, the factor $1 - p$ is near 1, so roughly $\mathrm{Var}(g) \approx p\,\|\nabla \log \pi_\theta(y^*)\|^2$. In other words, when the model has low chance to generate correct answers, the gradient estimator is almost always zero but occasionally a large spike, yielding extremely high variance relative to its mean. This high variance makes the training updates very unstable, as gradient estimates will fluctuate widely between zero and large values. We conclude that, with a binary reward, most of the time the model receives no learning signal, and when it does, the signal is an outsized jump. $\square$

**Theorem 4.2** (Exploration-Exploitation Trade-off and Hallucination Risk). *Given the high-variance gradient under a binary reward, maximizing the expected binary reward objective $\mathcal{J}(\theta)$ will enforce an implicit constraint that the entropy $H_\theta(x)$ remains sufficiently large. Let $\epsilon > 0$ be a small constant reflecting minimum acceptable exploration probability. To avoid stagnation at incorrect local optima without gradient feedback, the optimal policy update rule must satisfy: $H_\theta(x) \geq H_{min}(\epsilon) > 0$.*

*Proof.* The goal of policy model is to increase the probability of producing correct answers. In early stages or under uncertain knowledge (i.e., $p$ is small), gradient updates are extremely noisy, often zero or very large (Theorem 4.1). To reliably find rewarding outputs, the agent must ensure it samples broadly from its policy distribution. Given $\epsilon > 0$ the minimal exploration threshold of ensuring a non-negligible probability of sampling potentially correct outputs, the exploration constraint is:

$$\pi_\theta(y|x) \geq \epsilon, \quad \forall y \in Y_{\text{unexplored}}, \tag{6}$$

where $Y_{\text{unexplored}}$ is a subset of unexplored outputs that have unknown correctness. Maintaining such a constraint implies a sufficiently high entropy $H_\theta(x)$, because low entropy distributions concentrate their mass on few outputs, increasing the risk that these outputs yield zero reward consistently:

$$H_\theta(x) = -\sum \pi(y|x) \log \pi(y|x) \geq H_{\min}(\epsilon) > 0. \tag{7}$$

The necessity for exploration to ensure non-stagnation at zero-reward local minima introduces the exploration-exploitation trade-off. On one hand, the entropy must remain relatively high to explore potential answers, increasing the probability of hallucinated outputs. On the other hand, maximizing immediate reward requires concentrating distribution on already known rewarding answers. $\square$

**Theorem 4.3** (Local Optima and Incorrect Convergence). *Binary reward modeling admits spurious local optima where the policy produces incorrect answers with high confidence and receives zero reward, yet has no gradient incentive to change. Any policy that deterministically outputs a particular wrong answer is a stationary point of the optimization, i.e., once the model heavily commits to a wrong answer, the policy gradient provides no immediate push away from that policy.*

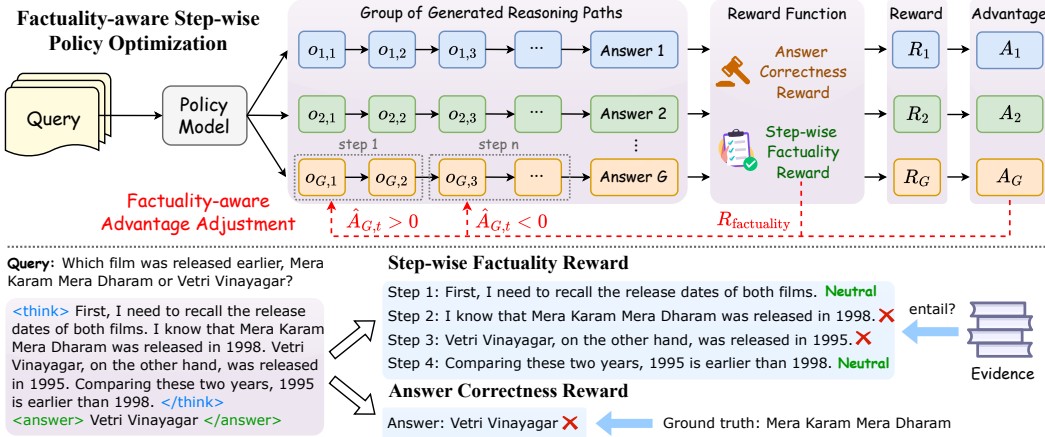

Figure 3: Overall framework of our proposed FSPO approach.

*Proof.* Given the objective $\mathcal{J}(\theta) = \sum \mathcal{R}(y)\pi_\theta(y|x)$, it is maximized at $\mathcal{J} = 1$ when the policy outputs a correct answer with probability 1. However, when a policy is certain but wrong for some incorrect answers $y_{\text{false}}$, the probability $\pi_\theta(y_{\text{false}}|x) = 1$ (so $\pi_\theta(y^*|x) = 0$ for all correct answers $Y^*$). For this policy, the expected reward $\mathcal{J}(\theta) = 0$. Importantly, the gradient $\nabla_\theta \mathcal{J}(\theta)$ is zero as well, since $\nabla_\theta \mathcal{J} = \mathbb{E}[\mathcal{R}(y)\nabla \log \pi_\theta(y|x)]$ and $\mathcal{R}(y)$ is always 0 under this policy (only generating incorrect answers). Thus, $\theta$ is a stationary point (the gradient is zero and no update is favored).

In a strict mathematical sense, the policy of always outputting a particular wrong answer is actually a local minimum of the objective $\mathcal{J}(\theta)$, since any small increase in the probability of a correct answer would increase $\mathcal{J}(\theta)$ slightly. However, the surface around it is a broad, flat plateau at reward zero for every direction not immediately introducing the correct answer. Since REINFORCE updates are driven solely by sampled positive rewards, the algorithm has no gradient signal to push the parameters off this plateau. In effect, the training has converged to a policy that outputs a wrong answer with full confidence, and because it never explores the correct answer, it gets no feedback. We call such states spurious local optima: the model is confidently wrong and training fails to progress. □

## 5 Approach

To mitigate the hallucination issue during large-scale RL training, we propose the **F**actuality-aware **S**tep-wise **P**olicy **O**ptimization (**FSPO**) algorithm. As shown in Figure 3, by integrating step-wise factuality signals into the policy optimization process, our approach encourages the model to generate more factual and faithful reasoning steps.

### 5.1 Reward Function Design

The core of reinforcement learning is to design appropriate reward functions that encourage high-quality reasoning trajectories and penalizes low-quality ones, guiding the policy model towards better responses using the RL algorithm. In our method, we introduce two rewards, i.e., answer correctness reward and step-wise factuality reward.

**Step-wise Factuality Reward.** Typically, factuality is mainly related to world knowledge, so we focus on complex knowledge-intensive tasks in this work, e.g., multi-hop question answering [41]. Specifically, for an input question $x$, we assume that it can be associated with knowledge snippets in a source such as Wikipedia, denoted by $\mathcal{K}$, which can be called *evidence*. We first split the reasoning text into multiple sentences as steps, denoted by $\{z_1, ..., z_N\}$, and then employ an LLM to determine whether the evidence $\mathcal{K}$ can entail each sentence $z_j$ to assign a step-wise factuality reward as:

$$\mathcal{R}_{\text{factuality}}(z_j) = \begin{cases} 1, & \text{if the sentence } z_j \text{ can be entailed from the evidence } \mathcal{K} \\ 0, & \text{if the sentence } z_j \text{ is neutral to the evidence } \mathcal{K} \\ -1, & \text{if the sentence } z_j \text{ contradicts the evidence } \mathcal{K} \end{cases} \quad (8)$$

We observe that the policy model might generate texts unrelated to the question in the reasoning process such as connective phrases or exploratory sentences (e.g., "Aha", "Wait"), which can be regarded as neutral to the evidence.

**Answer Correctness Reward.** In most existing research [9, 38, 33], answer correctness reward is usually adopted due to its significant scalability and simplicity. Specially, given an input prompt $x$, the policy model generates an output $y$, which includes intermediate reasoning texts followed by the final answer. We use a rule-based reward function to evaluate an answer's correctness as follows:

$$\mathcal{R}_{\text{answer}}(y) = \begin{cases} 1, & \text{if the final answer fully matches the ground truth} \\ 0, & \text{if the final answer does not match the ground truth} \end{cases} \tag{9}$$

We simplify the answer reward design aiming to alleviate reward hacking and promote more diverse problem-solving behaviors based on mere outcome feedback.

To reduce the effect of learning from sparse outcome reward, the final reward for an output $y$ consists of both answer correctness and step-wise factuality rewards, which can provide denser feedback and mitigate the *high-variance gradient* issue (Theorem 4.1) in typical RL binary reward modeling:

$$\mathcal{R}_{\text{final}}(y) = \mathcal{R}_{\text{answer}}(y) + \frac{1}{N} \sum_{j=1}^{N} \mathcal{R}_{\text{factuality}}(z_j). \tag{10}$$

## 5.2 Factuality-Aware Policy Optimization

For online policy optimization, we adopt Group Relative Policy Optimization (GRPO) [30], which samples a group of outputs $\{y_1, ..., y_G\}$ for each input $x$ and then obtains their rewards $\{\mathcal{R}_1, ..., \mathcal{R}_G\}$ to compute the advantage $A_i$ for $y_i$ as $A_i = (\mathcal{R}_i - \text{mean}(\{\mathcal{R}_1, ..., \mathcal{R}_G\}))/\text{std}(\{\mathcal{R}_1, ..., \mathcal{R}_G\})$.

**Factuality-Aware Advantage Adjustment.** Most existing work assigns the same advantage (derived from the overall response) to all tokens [30, 38], which is suboptimal and does not differentiate between them. We argue that, regardless of whether the final answer is correct, factually correct reasoning tokens should be encouraged, while incorrect reasoning tokens should be penalized. Therefore, we adjust the advantage of each token $o_{i,t} \in z_j$ for the output $y_i$ as follows:

$$\hat{A}_{i,t} = \begin{cases} A_i, & \text{if } A_i > 0 \wedge \mathcal{R}_{\text{factuality}}(z_j) = 1 \text{ or } A_i < 0 \wedge \mathcal{R}_{\text{factuality}}(z_j) = -1 \\ -A_i, & \text{if } A_i > 0 \wedge \mathcal{R}_{\text{factuality}}(z_j) = -1 \text{ or } A_i < 0 \wedge \mathcal{R}_{\text{factuality}}(z_j) = 1 \\ A_i, & \text{if } A_i = 0 \text{ or } \mathcal{R}_{\text{factuality}}(z_j) = 0 \end{cases} \tag{11}$$

In this manner, each token $o_{i,t}$ is re-weighted by the factuality-aware advantage $\hat{A}_{i,t}$. Whenever the generated reasoning contains at least one sentence that is contradicted or entailed by the evidence ($\mathcal{R}_{\text{factuality}} = 1, -1$), the advantages for tokens inside that sentence are not zero, even if the overall answer is still wrong. Consequently, the gradient is strictly non-zero, supplying a descent (or ascent) direction that pushes probability mass away from factually-incorrect tokens and toward factually-verified ones and enabling gradient-based learning to leave the *spurious local optima* (Theorem 4.3). Based on the adjusted advantage, we adopt the objective in Eq. 4 to optimize the policy model:

$$\mathcal{J}_{\text{FSPO}}(\theta) = \mathbb{E}_{y \sim \pi(\cdot|x)}$$
$$\frac{1}{G} \sum_{i=1}^{G} \frac{1}{|y_i|} \sum_{t=1}^{|y|} \left\{ \min \left[ \frac{\pi_\theta(o_{i,t}|x, y_{i,<t})}{\pi_{\theta_{\text{old}}}(o_{i,t}|x, y_{i,<t})} \hat{A}_{i,t}, \text{clip} \left( \frac{\pi_\theta(o_{i,t}|x, y_{i,<t})}{\pi_{\theta_{\text{old}}}(o_{i,t}|x, y_{i,<t})}, 1 - \epsilon, 1 + \epsilon \right) \hat{A}_{i,t} \right] \right\}. \tag{12}$$

# 6 Experiments

In this section, we present our experimental setup, the main results, and our analysis of the results.

## 6.1 Experimental Setup

**Datasets and Metrics.** To perform FSPO, we adopt a challenging subset of HotpotQA [41] and 2WikiMultiHopQA [11] from previous work [33] and randomly select 2K examples for training

(including questions, answers, and corresponding Wikipedia segments). Besides, to improve the complex reasoning capabilities of LLMs, we incorporate the SimpleRL dataset [45] with 8K mathematical problems for vanilla RL training. For hallucination evaluation, we adopt three widely-used benchmarks, i.e., TruthfulQA [20], HaluEval [19], and HalluQA [4]. For HaluEval, we adopt the QA subset and report the accuracy with which the model can discriminate correct and hallucinated answers. For TruthfulQA and HalluQA, we report the ratio of outputs that the judge model predicts as truthful. We employ GSM8K [5], MATH-500 [10], AIME 2024 [22], and AIME 2025 [23] to evaluate mathematical reasoning performance and report the Pass@1 accuracy results in these datasets. For more details, refer to Appendix C.

**Baselines.** We employ Qwen2.5-7B-Base/Instruct [39] and Llama3.1-8B-Instruct [8] as backbone models for our approach. We compare our model against three groups of baselines:

• **API-based Models** include DeepSeek-V3 [21], DeepSeek-R1 [9], OpenAI GPT-4o, and GPT-o1.

• **Reasoning Models** consist of QwQ-32B [28], Distill-Qwen-7B/14B/32B, and Distill-Llama-8B [9], which are fine-tuned on the long CoT data (800K samples) synthesized from DeepSeek-R1.

• **Open-source Models** involve Qwen2.5-7B-Base/Instruct and Llama3.1-8B-Instruct, which are direct comparisons and can be easily accessed through their open-source weights.

**Implementation Details.** We train our model using the verl [31] framework. Specifically, during training, we use a batch size of 8, generate 8 rollouts for each prompt, set a maximum prompt and response length of 2,048 tokens, and train using a mini-batch size of 1,024. We train our model on the mixture dataset of HotpotQA subset and SimpleRL for one epoch with a constant learning rate of 4e-7 and temperature parameter of 1.0. The coefficient for KL loss is set to 1e-3 and the clip ratio is 0.2. We adopt the HHEM-2.1 model [14] to verify step-wise factuality. During evaluation, we set the sampling temperature to 1.0. We present prompt templates throughout this work in Appendix F.

## 6.2 Main Results

The experimental results presented in Table 1 demonstrate the performance of our proposed method, FSPO, across various reasoning and hallucination benchmarks compared to multiple competitive models, including API-based, reasoning, and open-source alternatives. On hallucination benchmarks, FSPO clearly outperforms all open-source, reasoning and even some API-based models. Specifically, FSPO based on Qwen2.5-7B-Base achieves the highest scores among all open-source and reasoning models on hallucination datasets, clearly demonstrating the effectiveness of our training approach in substantially reducing hallucinations and enhancing factual correctness.

On reasoning benchmarks, FSPO achieves superior results within the open-source category, notably surpassing other base models like Qwen2.5-7B-Instruct and Llama3.1-8B-Instruct by significant margins (e.g., GSM8K 89.5% vs. 73.2% and 77.5%, respectively). FSPO even surpasses some reasoning models on GSM8K, and maintains competitive performance on MATH-500. This shows that our FSPO method can enhance reasoning capabilities while reducing hallucinations.

In terms of advanced reasoning tasks, FSPO delivers competitive outcomes, especially on AIME 2024 and 2025, with scores of 20.0% and 13.3% respectively. These scores notably surpass smaller base models, although trailing the reasoning and much larger API-based models. Such results underscore FSPO's robust mathematical reasoning capabilities, showcasing the efficacy of our specialized fine-tuning methods in elevating performance on complex mathematical reasoning tasks despite the base model's comparatively modest size.

## 6.3 Further Analysis

We further conduct detailed analysis on several key aspects of our approach. Due to similar findings, the following analysis is focused on MATH-500 and HaluEval-QA with Qwen2.5-7B-Base.

**Ablation Analysis.** To validate the effectiveness of our approach, we ablate its key design elements. We compare FSPO to two variants: 1) *GRPO* trains the model on the mixture dataset using typical GRPO with only answer reward in Eq. 9; and 2) *GRPO w/ factuality* incorporates both answer and factuality rewards in Eq. 10. The results from Figure 4 further support the effectiveness of FSPO,

Table 1: Evaluation results of closed-source, distilled, and open-source models across four reasoning benchmarks and three hallucination benchmarks. The **bold** and underlined scores denote the best and second best results among open-source models and our trained models.

| Model | GSM8K | MATH 500 | AIME24 (Pass@1) | AIME25 (Pass@1) | TruthfulQA↑ | HaluEval-QA↑ | HalluQA↑ |
|---|---|---|---|---|---|---|---|
| **API-based Models** | | | | | | | |
| DeepSeek-V3 | 89.3 | 90.2 | 39.2 | 26.6 | 53.0 | 70.6 | 59.1 |
| DeepSeek-R1 | 92.0 | 97.3 | 79.8 | 66.7 | 45.2 | 51.8 | 51.1 |
| GPT-4o-0513 | 90.8 | 74.6 | 9.3 | 13.3 | 59.0 | 62.6 | 53.1 |
| GPT-o1-1217 | 92.3 | 96.4 | 79.2 | 76.7 | 62.4 | 70.2 | 56.0 |
| **Reasoning Models** | | | | | | | |
| QwQ-32B | 88.6 | 89.8 | 79.5 | 56.7 | 23.2 | 29.6 | 23.7 |
| R1-Distill-Qwen-32B | 87.4 | 94.3 | 72.6 | 53.3 | 19.7 | 33.5 | 26.9 |
| R1-Distill-Qwen-14B | 85.1 | 93.9 | 69.7 | 50.0 | 18.2 | 24.8 | 20.2 |
| R1-Distill-Qwen-7B | 84.3 | 92.8 | 55.5 | 33.3 | 6.9 | 11.6 | 3.1 |
| R1-Distill-Llama-8B | 82.1 | 89.1 | 50.4 | 26.7 | 8.8 | 14.6 | 4.6 |
| **Open-source Models** | | | | | | | |
| Qwen2.5-7B-Base | 65.2 | 35.7 | 3.3 | 3.3 | 38.2 | 48.0 | 39.5 |
| Qwen2.5-7B-Instruct | 73.2 | 51.6 | 6.7 | 3.3 | 36.7 | 38.4 | 32.0 |
| Llama3.1-8B-Instruct | 77.5 | 33.1 | 6.7 | 0.0 | 26.4 | 36.7 | 12.2 |
| FSPO (Qwen-Base) | **89.5** | **75.5** | 16.7 | **13.3** | **58.4** | **83.0** | **52.0** |
| FSPO (Qwen-Instruct) | 89.4 | 74.7 | **20.0** | **13.3** | 54.0 | 64.7 | 50.0 |
| FSPO (Llama-Instruct) | 86.2 | 68.3 | 13.3 | 6.7 | 41.1 | 67.1 | 42.0 |

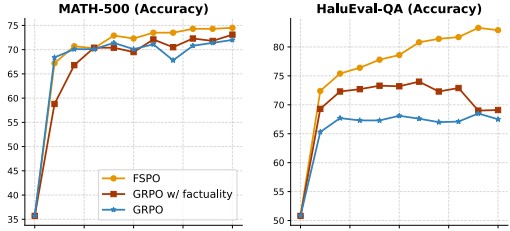

Figure 4: Pass@1 accuracy over training steps for ablation analysis.

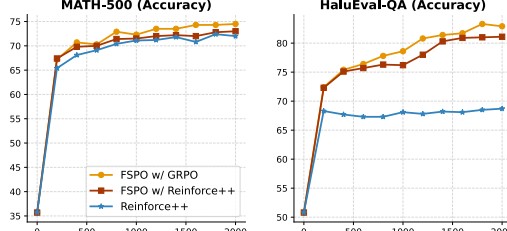

Figure 5: Pass@1 accuracy over training steps with different RL algorithms.

clearly outperforming baseline methods GRPO and GRPO with factuality scores across training steps on both MATH-500 and HaluEval-QA. FSPO maintains consistently superior reasoning performance while achieving continuous improvement in reducing hallucinations, indicating robust generalization capabilities and the practical benefits of our proposed method.

**RL Algorithm Analysis.** We further adapt FSPO to the Reinforce++ algorithm [12] for validating the adaptability and effectiveness of FSPO beyond GRPO. As shown in Figure 5, FSPO with Reinforce++ exhibits significant improvement over Reinforce++ alone, closely following the performance of FSPO with GRPO. Moreover, FSPO with GRPO consistently achieves higher accuracy than the Reinforce++ baselines across training steps on both the MATH-500 and HaluEval-QA benchmarks. These findings indicate that our proposed FSPO approach generalizes well to other reinforcement learning algorithms, enhancing their effectiveness and stability in solving complex reasoning and reducing hallucinations.

**Number of Training Samples.** To investigate the model performance under varying numbers of training samples, we randomly sample 1K, 2K, 4K, and 8K samples from the HotpotQA subset [33], with the results shown in Figure 6. On MATH-500, the accuracies are similar from 1K to 4K samples while diminishing with 8K samples, which might be because more factual QA data lower the math reasoning performance. In contrast, on the HaluEval-QA task, significant accuracy improvement continues up to 2K samples, with 4K and 8K samples achieving mediocre results. This demonstrates that our approach can significantly reduce hallucinations using only a small amount of data, without compromising the model's capacity for complex reasoning.

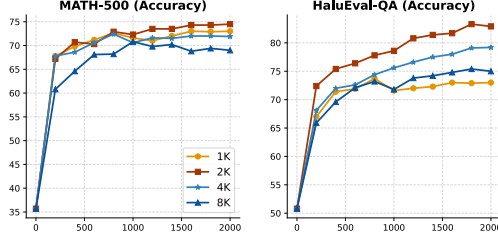 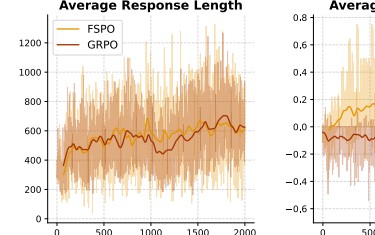 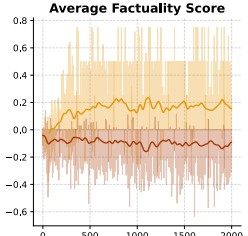

Figure 6: Pass@1 accuracy over training steps with varying number of training samples.

Figure 7: Average response length and factuality score for FSPO and GRPO.

**Factuality Improvement Analysis.** We further explore the benefit of incorporating step-wise factuality reward in our approach. The results in Figure 7 illustrate that both FSPO and GRPO generate responses of similar lengths. However, as our proposed FSPO approach explicitly optimizes for factual correctness during intermediate reasoning steps, it achieves significantly higher factuality scores compared to GRPO. This suggests that FSPO effectively enhances model reliability and factual accuracy without compromising the quality of the generated content.

# 7 Related Work

**Large Reasoning Models.** Recent advancements in large reasoning models have marked a significant shift toward explicitly improving complex reasoning, planning, and problem-solving capabilities [34]. OpenAI's o1 [15] exemplifies the integration of RL with long CoT reasoning, enabling the model to iteratively refine its reasoning process before producing an answer. This approach significantly enhances its capability in complex problem-solving scenarios, particularly in mathematical reasoning and algorithmic tasks. DeepSeek-R1 [9] and Kimi k1.5 [17] similarly emphasize RL-based reasoning but introduce an open and flexible training pipeline. QwQ model [28] underscores the potential of smaller models when optimized specifically for reasoning tasks. Increasing efforts are devoted to reproducing prominent reasoning performance of these models [43, 27, 3]. However, existing research has predominantly concentrated on enhancing the reasoning capabilities of models, while overlooking the issue of hallucinations emerging within these models.

**Hallucination.** Hallucination has been a fundamental challenge in large language models (LLMs), receiving extensive attention in existing literature [13, 16, 46]. To investigate and detect hallucinations in LLMs, in scenarios where LLMs provide access to their internal states, researchers can closely examine their internals to uncover underlying mechanisms leading to hallucinations [36, 44]. Conversely, when the internal states of LLMs are inaccessible and they can only be queried through external API calls, researchers typically approach hallucination detection by examining correlations between input prompts and generated responses [24, 42]. To mitigate hallucinations, existing studies propose various strategies. During the pre-training phase, mitigation efforts typically revolve around curating and cleaning datasets [7, 35]. Following pre-training, fine-tuning methods such as supervised fine-tuning [37] and reinforcement learning with human feedback [26] are commonly employed to further mitigate hallucinations. Additionally, post-processing mitigation involves verifying the factual correctness of generated content using either the LLM itself [25] or external knowledge-based verification methods [1]. Our work is an early effort focusing on analyzing and mitigating hallucinations in large reasoning models.

# 8 Conclusion

In this paper, we have highlighted and theoretically explained a critical limitation associated with RL-based fine-tuning of LLMs, specifically the increased incidence of hallucinations in generated reasoning content. Our analysis attributed this issue to unstable training dynamics driven by high-gradient variance, entropy-induced randomness, and convergence toward spurious local optima inherent in traditional outcome-based RL training. To address these challenges, we introduced FSPO, a novel factuality-aware RL algorithm that integrates step-wise factual verification, providing a denser and more accurate feedback mechanism during training. Comprehensive experimental evaluations

demonstrate that FSPO significantly mitigates hallucinations across multiple benchmarks without compromising the models' reasoning capabilities.

## 9 Acknowledgments

This research is supported by the National Research Foundation Singapore under its AI Singapore Programme (Award Number: AISG3-RP-2022-030).

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

## Technical Appendices and Supplementary Material

## A Limitations

Our theoretical analysis is mainly focused on the prevalent and simplified binary reward modeling. Future work can extend it to any arbitrary bounded real rewards. The scope of our experiments is constrained by the substantial computational cost of much larger models (e.g., 14B, 32B). Future work can explore applying our approach to larger models.

## B Broader Impact

Our proposed FSPO algorithm has the potential to significantly improve the trustworthiness and reliability of LLMs in critical applications, such as education, healthcare, and scientific research, by reducing factual hallucinations during complex reasoning. However, the same technology may also pose negative societal risks if misused, for example, it could be applied to train persuasive yet selectively factual systems in domains such as political propaganda or misinformation campaigns, where step-wise coherence may mask underlying biases.

## C Datasets

To reduce hallucinations while improving the complex reasoning capabilities of LLMs, we mix two datasets for fine-tuning base models, i.e., a subset of HotpotQA [33] and SimpleRL [45]. Previous study [33] construct a subset from HotpotQA [41] and 2WikiMultiHopQA [11] with varying difficulty levels including 4761 and 3737 samples from HotpotQA and 2WikiMultiHopQA, respectively. Each sample includes the question, ground-truth answer, and the corresponding segments from Wikipedia. We randomly select 2K samples from this subset for conducting our proposed FSPO approach. SimpleRL consists of 8523 mathematical samples, which are used to perform traditional RL fine-tuning based on GRPO in our training. For hallucination datasets, TruthfulQA [20], HaluEval-QA [19], and HalluQA [4] contain 817, 10000, 450 samples, respectively. HalluQA is a Chinese dataset while the others are English datasets. For mathematical reasoning datasets, GSM-8K [5] and MATH-500 [10] contains 1319 and 500 samples, respectively. There are 30 samples in both AIME 2024 [22] and AIME 2025 [22]. These datasets are in MIT or Apache-2.0 license.

## D Training Infrastructure

We train our model on 4×H100 80G GPU cards for about 24 hours each time. We train our model using the verl [31] framework. During training, we use a train batch size of 8, generate 8 rollouts for each prompt, set a maximum prompt and response length of 2,048 tokens, and train using a mini-batch size of 1,024. We use a constant learning rate of 4e-7 and temperature parameter of 1.0. The coefficient for KL loss is set to 1e-3 and the clip ratio is 0.2. During evaluation, we set the sampling temperature to 1.0 for our models and all baselines.

## E Preliminary Experiments

To demystify the question of "*To what extent do reasoning models maintain factuality after RL fine-tuning?*", we conduct preliminary experiments on three hallucination benchmarks with twelve LLMs. The results show that after large-scale training with RL or long CoT data, models demonstrate significantly higher degree of hallucinations. To investigate the underlying reasons, we randomly select 200 samples (i.e., examples that DeepSeek-V3 answers correctly but DeepSeek-R1 does not) from HaluEval-QA and manually check their hallucination reasons. Specifically, the hallucination sources within our experiments can be roughly categorized into four types:

- *Intermediate Reasoning Error*: This is the most common source where the intermediate reasoning process is incorrect, which naturally results in wrong answers.
- *Reasoning and Answer Inconsistency*: This is the second most common source for which the intermediate reasoning process is correct but the final answer is wrong.

Table 2: Prompt for base and instruct models in mathematical reasoning tasks.

| Model Type | Prompt |
| --- | --- |
| Base model | A conversation between User and Assistant. The user asks a question, and the assistant solves it. The assistant first thinks about the reasoning process in the mind and then provides the user with the final answer. The reasoning process and answer are enclosed within <think> </think> and <answer> </answer> tags, respectively. For example, <think> reasoning process here </think> <answer> answer here </answer>. The Assistant shows the reasoning process within <think> </think> tags, and ONLY return the FINAL AN-SWER within <answer> </answer> tags. For example: <answer> 12 </answer>. \n\n User: {question} \n Assistant: Let me solve this step by step. <think> |
| Instruct model | You are a helpful assistant. Given a question, you need to first think about the reasoning process in the mind and then provide the final answer. The reasoning process and answer are enclosed within <think> </think> and <answer> </answer> tags, respectively. For example, <think> reasoning process here </think> <answer> answer here </answer>. You must show the reasoning process within <think> </think> tags, and ONLY return the FINAL ANSWER within <answer> </answer> tags. For example: <answer> 12 </answer>. |

Table 3: Prompt for base and instruct models in question answering tasks.

| Model Type | Prompt |
| --- | --- |
| Base model | A conversation between User and Assistant. The user asks a question, and the assistant solves it. The assistant first thinks about the reasoning process in the mind and then provides the user with the final answer. The reasoning process and answer are enclosed within <think> </think> and <answer> </answer> tags, respectively. For example, <think> reasoning process here </think> <answer> answer here </answer>. The Assistant shows the reasoning process within <think> </think> tags, and ONLY return the FINAL AN-SWER within <answer> </answer> tags. For example: <answer> Kim Marton </answer>. \n\n User: {question} \n Assistant: Let me solve this step by step. <think> |
| Instruct model | You are a helpful assistant. Given a question, you need to first think about the reasoning process in the mind and then provide the final answer. The reasoning process and answer are enclosed within <think> </think> and <answer> </answer> tags, respectively. For example, <think> reasoning process here </think> <answer> answer here </answer>. You must show the reasoning process within <think> </think> tags, and ONLY return the FINAL ANSWER within <answer> </answer> tags. For example: <answer> Kim Marton </answer>. |

- *Overthink*: This occurs when the intermediate reasoning process contains correct segments that can lead to the right answers but the model still continues to reason further, finally getting the wrong answers. This phenomenon is called "*overthink*", investigated in extensive studies [6].

- *Others*: This includes the remaining cases, such as the output text is incomplete or a mismatch between generated answers and ground truth.

# F    Prompt Templates

We present our prompts used during training and evaluation in Table 2, Table 3, and Table 4.

Table 4: Prompt for base and instruct models in HaluEval-QA dataset.

| Model Type | Prompt |
|---|---|
| Base model | A conversation between User and Assistant. The user asks a question, and the assistant solves it. The assistant first thinks about the reasoning process in the mind and then provides the user with the final answer. The reasoning process and answer are enclosed within <think> </think> and <answer> </answer> tags, respectively. For example, <think> reasoning process here </think> <answer> answer here </answer>. Now the user asks a question and provides two candidate answers, the assistant needs to determine which answer is correct. The Assistant shows the reasoning process within <think> </think> tags, and ONLY return the FINAL ANSWER within <answer> </answer> tags. For example: <answer> Kim Marton </answer>. \n\n User: #Question#: {question} \n #Candidate Answer 1#: {candidate_answer_1} \n #Candidate Answer 2#: {candidate_answer_2} \n Assistant: Let me solve this step by step. <think> |
| Instruct model | You are a helpful assistant. Given a question, you need to first think about the reasoning process in the mind and then provide the final answer. The reasoning process and answer are enclosed within <think> </think> and <answer> </answer> tags, respectively. For example, <think> reasoning process here </think> <answer> answer here </answer>. Now the user asks a question and provides two candidate answers, you need to determine which answer is correct. You must show the reasoning process within <think> </think> tags, and ONLY return the FINAL ANSWER within <answer> </answer> tags, such as <answer> Kim Marton </answer>. |

