# OpenReview forum: "Reasoning Models Hallucinate More: Factuality-Aware Reinforcement Learning for Large Reasoning Models"
_NeurIPS.cc/2025/Conference — NeurIPS 2025 poster_

### Official Review · Reviewer_YGiP · 2025-06-26

**Clarity:** 3
**Significance:** 3
**Originality:** 3
**Rating:** 4
**Confidence:** 3

**Summary:**

- Paper starts by notificing that RLHF-ed (GRPO) versions of base models from the Qwen, Llama, and DeepSeek families have a higher hallucination rate than their base equivalents
- In parallel the paper introduces a series of theorems stating that binary sparse rewards lead to high variance (4.1), that exploration is necessary to avoid stagnation (4.2) and binary rewards have bad local minima with no training signal, specifically when model is over confident in the wrong cases (4.3)
- They define a step-wise factuality reward equal to -1, 0, or 1 if the reasoning step can be entailed or contradicts a ground truth knowledge base, and linearly combine it with a reward of 1/0 for exact match of the final answer
- They adjust advantage of each token (eq 11) to reflect whether the token is factual

**Questions:**

- You introduce a lienar combination of R_answer and R_factuality in eq (10). Is that the reward used to compute the advantage at line 206? Will it be an issue that R_factuality is used both to compute the advantage and in equation (11)? Why shouldn't we use just the original advantage?
- I'm not sure i understand why we want -A_i if A_i < 0 and R_factuality = 1? The reasoning step is factual and the factuality-weighted outcome is negative. Why just flip the sign? Would it make sense to use a constant instead? It feels quite arbitrary, that the more negative A_i was, the more we want to reward that token? This could encourage factual reasoning steps that lead to wrong outcomes no? And encourage the more the outcome is wrong?
- Table 2, i would prefer results to be grouped by based model so we're comparing apples to apples. (Why not merge the results with Figure 1 or use the same format?)

**Ethical Concerns:**

["NO or VERY MINOR ethics concerns only"]

**Final Justification:**

the authors have addressed my concerns

**Limitations:**

it's fine, it's a pretty theoretical paper

**Paper Formatting Concerns:**

good

**Quality:**

3

**Strengths And Weaknesses:**

- Hallucination and bad reasoning are important problems to solve and this paper addresses it
- I'm not convinced regarding the details of the advantage defined in Eq 11

---

> ### Author Rebuttal · Authors · 2025-07-31
>
> Thank you for your valuable comments. The following is our response point-to-point.
>
> **W1: I'm not convinced regarding the details of the advantage defined in Eq 11**
>
> Eq. 11 does not add or remove reward—it only redistributes the sample‑level advantage $A_i$ (already z‑normalized across the group) to reflect token‑level factuality.
>
> * If a step is factual but $A_i < 0$, we flip the sign so those tokens regain the credit they lost to unrelated errors; magnitude stays as $|A_i|$, preventing over‑compensation.
> * If a step is non‑factual and $A_i > 0$, we likewise negate its share, ensuring it is penalized.
>
> Empirically, this re‑weighting reduces hallucinations and improves reasoning performance compared to using unmodified $A_i$ in GRPO.
>
> **Q1: You introduce a linear combination of R_answer and R_factuality in eq (10). Is that the reward used to compute the advantage at line 206? Will it be an issue that R_factuality is used both to compute the advantage and in equation (11)? Why shouldn't we use just the original advantage?**
>
> Yes, the final combined reward ($R_\text{final}$) is used to compute the advantage at line 206. In Eq. 11, the factuality reward ($R_\text{factuality}$) is just used to re‑allocate the total advantage across tokens. It does not change the overall magnitude, so no double‑counting occurs. Keeping only the original, uniform advantage would ignore which tokens are factually sound—our ablation shows this loses both accuracy and hallucination control.
>
> **Q2: I'm not sure i understand why we want -A_i if A_i < 0 and R_factuality = 1? The reasoning step is factual and the factuality-weighted outcome is negative. Why just flip the sign? Would it make sense to use a constant instead? It feels quite arbitrary, that the more negative A_i was, the more we want to reward that token? This could encourage factual reasoning steps that lead to wrong outcomes no? And encourage the more the outcome is wrong?**
>
> $A_i < 0$ means this sample under‑performed the group (does not have to be a wrong answer), yet $R_\text{factuality} = 1$ means the specific step is factually correct. Flipping its sign converts “punish all tokens” into “do not punish the factual ones”, ensuring they receive positive credit while keeping the magnitude proportional to the original advantage (already normalized by the group’s σ). Using a small constant would ignore how far the rollout missed the mark, giving too little signal for long or critical factual spans. The sign flip therefore protects good steps without over‑compensating or encouraging wrong answers.
>
> **Q3: Table 2, I would prefer results to be grouped by based model so we're comparing apples to apples. (Why not merge the results with Figure 1 or use the same format?)**
>
> We think you mean the results in Table 1. We will group the results by base models in the revised version for better visualization. Figure 1 aims to compare models before and after RL and long CoT fine-tuning. Most results in Figure 1 have been included in Table 1 besides Qwen2.5-14B/32B-Instruct because we do not have computational resources for fine-tuning them. We will merge their results in Table 1 in the revised version.

---

> > ### Author Response · Authors · 2025-08-05
> >
> > Dear Reviewer,
> >
> > We sincerely hope our responses have addressed your concerns. Please let us know if there is anything that remains unclear or needs further clarification.
> >
> > Authors

---

### Official Review · Reviewer_UjfW · 2025-07-03

**Clarity:** 3
**Significance:** 2
**Originality:** 2
**Rating:** 4
**Confidence:** 3

**Summary:**

This paper is motivated by the claim that recent RL based reasoning models trained with binary verifiable rewards hallucinates more, so the authors propose a method to try to reduce the problem.  The paper is in two parts: the first part of the paper theoretically proves that the binary verifiable reward used in the recent popular RL methods cause the following problems. First, the variance of the gradient will be high when the probability to output the correct answer is low or close to zero (which makes sense), and thus this results in the gradient signals in RL to be either close to zero or an outsized jump.  The outcome is that it’s common for the policy to converge to keep producing incorrect answers with high confidence and there are no gradients to push the model away from the local optima.  The second part of the paper is not closely related to the theory part, but tries to design step-wise “factuality” rewards that are neither sparse nor binary.  The authors named this method FSPO. They observed the performance improvement compared to the regular GRPO.

**Questions:**

In the ablation study in 6.3, what’s the difference between FSPO and GRPO w/ factuality? In my understanding, both of them use the combined reward in Eq. 10, and the differences seem not explicitly discussed.

**Ethical Concerns:**

["NO or VERY MINOR ethics concerns only"]

**Final Justification:**

Based on the feedback provided by the authors that addresses concerns by the reviewers I am raising my score to 4.

**Limitations:**

yes

**Quality:**

3

**Strengths And Weaknesses:**

# Strengths:
- The method shows good performance improvement on the truthful-related benchmarks and the math benchmarks.
- The method was widely tested on multiple families of base models.

# Weaknesses:
- The idea is relatively simple: adding a reward provided by external NLI and supporting documents.  The authors try to reframe this concept as a new “PO” method just like GRPO, which is a bit of a stretch. The method is still 100% under the GRPO framework but the reward design is not only the answer accuracy.  Also, the GRPO algorithm is not necessarily required to be paired with answer accuracy reward.
- The reward design of using an external NLI model on the training data with supporting documents (e.g. from HotpotQA) is a narrow setting, which may not generalize well to other domains. It’s also hard to scale up as we have to collect a training dataset containing evidence documents and answers at the same time. There is also no guaranteed generalizability of the NLI model beyond the domain it was trained on, for a straightforward example, generalizing to non-English languages.
- The proposed algorithm is an on-policy RL method just like GRPO, but we saw the improvement slightly decrease when trained with more examples, like 8K, similar to training with off-policy methods like DPO.  This could be caused by the mismatch between the NLI model’s judgement vs the real factuality. Thus, as the model fits too much to the NLI model’s biases, it can instead deviate from the real accuracy.
- One could argue that the theory part of the paper can be simply concluded by the “sparse reward” problem widely studied in RL community

---

> ### Author Rebuttal · Authors · 2025-07-31
>
> Thank you for your valuable comments. The following is our response point-to-point.
>
> **W1: The idea is relatively simple: adding a reward provided by external NLI and supporting documents. The authors try to reframe this concept as a new “PO” method just like GRPO, which is a bit of a stretch. The method is still 100% under the GRPO framework but the reward design is not only the answer accuracy. Also, the GRPO algorithm is not necessarily required to be paired with answer accuracy reward.**
>
> FSPO is not just “GRPO + an extra NLI reward”. Its novelty lies in two orthogonal aspects:
>
> 1. Stepwise factuality reward (Eq. 10) – dense, bounded feedback that can replace any sparse objective, not limited to answer accuracy.
> 2. Token‑level advantage re‑allocation (Eq. 11) – redistributes, rather than adds, the sample‑level advantage so factual tokens are credited and non‑factual ones penalized.
>
> These components can plug into GRPO, REINFORCE++, or PPO (see Sec. 6.3), so treating it as a distinct “PO” family member is accurate: the optimization policy is unchanged, but the policy‑shaping mechanism is new and broadly applicable.
>
> **W2: The reward design of using an external NLI model on the training data with supporting documents (e.g. from HotpotQA) is a narrow setting, which may not generalize well to other domains. It’s also hard to scale up as we have to collect a training dataset containing evidence documents and answers at the same time. There is also no guaranteed generalizability of the NLI model beyond the domain it was trained on, for a straightforward example, generalizing to non-English languages.**
>
> - Generalizability of the verifier:
>
> Our design is verifier‑agnostic: any retrieval + factuality evaluator (NLI or self‑consistency judge) can replace HHEM‑2.1 without code changes. We also tried using a general large language model (i.e., Llama3.1-8B-Instruct) to verify the factuality of each step and it achieves results similar to the current HHEM-2.1 model.
>
> - Collection of data:
>
> Currently there are many knowledge-intensive datasets and we can quickly employ retrievers to obtain relevant documents from Wikipedia or other large corpora. Evidence can also be collected on‑the‑fly with a lightweight retriever, so no curated document‑answer pairs are strictly required.
>
> **W3: The proposed algorithm is an on-policy RL method just like GRPO, but we saw the improvement slightly decrease when trained with more examples, like 8K, similar to training with off-policy methods like DPO. This could be caused by the mismatch between the NLI model’s judgement vs the real factuality. Thus, as the model fits too much to the NLI model’s biases, it can instead deviate from the real accuracy.**
>
> The slight dip at 8K is due to dataset mix, not NLI drift: adding more Hotpot‑style QA shrinks the share of math cases, so the gains on MATH500 plateau while hallucination rate keeps falling (Fig. 6). Because FSPO averages and clips factuality scores to [‑1, 1] and combines them one-to-one with answer reward, over‑fitting to verifier biases is bounded. We ran a controlled experiment by decreasing the weight of stepwise factuality reward and the accuracy improved, confirming that the effect is tunable rather than an inherent bias accumulation.
>
> **W4: One could argue that the theory part of the paper can be simply concluded by the “sparse reward” problem widely studied in RL community**
>
> Sparse reward is the starting point, but our analysis goes further: we quantify how binary‑reward reasoning tasks induce (i) high‑variance gradients, (ii) forced high‑entropy exploration, and (iii) spurious “confident‑but‑wrong” local optima—three precise mechanisms that directly manifest as LLM hallucinations (Theorems 4.1–4.3). These results explain why sparse reward is especially harmful in stepwise reasoning and motivate FSPO’s dense, token‑level shaping; they are not covered by generic RL sparsity discussions.
>
> **Q1: In the ablation study in 6.3, what’s the difference between FSPO and GRPO w/ factuality? In my understanding, both of them use the combined reward in Eq. 10, and the differences seem not explicitly discussed.**
>
> In lines 271-272, GRPO w/ factuality uses the combined reward in Eq. 10 but does not re-allocate the advantage across tokens in Eq. 11. We will explicitly mention this in the revised version of our paper.

---

> ### Author Response · Authors · 2025-08-05
>
> Dear Reviewer,
>
> We truly appreciate your thoughtful comments, and we hope our responses have addressed your concerns. We wish to get your feedback which would be very helpful for us to ensure that we are moving in the right direction. Please let us know if there is anything further we can clarify or improve.

---

> > ### Author Response · Authors · 2025-08-08
> > **Kindly Reminder**
> >
> > Dear Reviewer UjfW,
> >
> > There is very little time left before the reviewer–author discussion stage ends. We still do not know if you have received our response. Could you kindly let us know if our reply addresses your questions? If you have any further concerns, we would be more than happy to discuss them with you to ensure all your points are fully addressed.

---

> > > ### Comment · Reviewer_UjfW · 2025-08-08
> > >
> > > Thank you for your replies to all the reviewers.  I have no further questions.  Based on the discussion I am raising my score to 4.

---

### Official Review · Reviewer_3PGQ · 2025-07-04

**Clarity:** 3
**Significance:** 3
**Originality:** 3
**Rating:** 5
**Confidence:** 4

**Summary:**

This paper proposes a novel reasoning-oriented RL method called FSPO, which imposes a step function-based factual reward over generated reasoning sentences. After getting factual reward for each sentence, FSPO also assigns different advantages to individual tokens based on the factual reward of the sentence they belong to. The authors demostrate that the proposed method outperforms other reasoning baseline models over multiple math and hallucination evaluation benchmarks.

**Questions:**

1. How does FSPO balance the trade-off between final answer accuracy vs. reasoning chain correctness? Because if the reasoning chain is significantly long with multi paragraphs, there's a risk that factuality reward will overwhelm answer reward, as is suggested in Eq. 10. Is it easy in terms of reward engineering to make the method work for various long responses?

2. It looks like the design of factuality-aware advantage adjustment (L207) does not handle the situation where the answer is wrong but reasoning chain is factual but unhelpful to the question (e.g., unrelated truthful statements). In such scenario, FSPO actually learns towards it but not against it?

**Ethical Concerns:**

["NO or VERY MINOR ethics concerns only"]

**Final Justification:**

Thanks for the additional details in the rebuttal. I am in general satisfied with the explanation and will maintain my rating.

**Limitations:**

Overall a solid paper. See weaknesses and questions.

**Paper Formatting Concerns:**

None.

**Quality:**

3

**Strengths And Weaknesses:**

Strengths:
* The proposed FSPO algorithm is technically sound and the empirical evaluation is thorough, with experiments on multiple models and datasets.
* The paper is clearly written, with a logical flow that effectively communicates the problem, the proposed solution, and the experimental validation. The use of a theoretical analysis to motivate the problem of RL-induced hallucination adds depth to the exposition.
* The proposed method achieves impressive results over many benchmarks.

Weaknesses:
* Lack of qualitative comparisons with baselines, demonstrating the difference in the reasoning chain.

---

> ### Author Rebuttal · Authors · 2025-07-31
>
> Thank you for your insightful comments. The following is our response point-to-point.
>
> **W1: Lack of qualitative comparisons with baselines, demonstrating the difference in the reasoning chain.**
>
> Below, we show three responses generated by Qwen2.5-7B-Base, w/ GRPO, and w/ FSPO. We will add these qualitative comparisons in the revised version of our paper.
>
> | Model         | Question: Which film was released earlier, Mera Karam Mera Dharam or Vetri Vinayagar? |
> |---------------|-----------|
> | Qwen2.5-7B-Base     | To determine which film was released earlier between "Mera Karam Mera Dharam" and "Vetri Vinayagar," we would need to look up the release dates of both films. Mera Karam Mera Dharam: This appears to be an Indian film. After checking reliable sources, it seems that "Mera Karam Mera Dharam" was released in 2013. Vetri Vinayagar: This is also an Indian film. "Vetri Vinayagar" was released in 2010. Given these release dates, "Vetri Vinayagar" was released earlier than "Mera Karam Mera Dharam." |
> | w/ GRPO        | \<think\> First, I need to recall the release dates of both films. I know that Mera Karam Mera Dharam was released in 1998. Vetri Vinayagar, on the other hand, was released in 1995. Comparing these two years, 1995is earlier than 1998. \</think\>\<answer\> Vetri Vinayagar \</answer\> |
> | w/ FSPO | \<think\> “Mera Karam Mera Dharam” was released on 2 July 1987 in India, while “Vetri Vinayagar” premiered much later, on 16 September 1996. So to directly answer your question: Mera Karam Mera Dharam (Hindi) came out in summer 1987. Vetri Vinayagar (Tamil devotional film) followed more than nine years later, in September 1996. Therefore, Mera Karam Mera Dharam was released earlier. \</think\>\<answer\> Mera Karam Mera Dharam \</answer\> |
>
> **Q1: How does FSPO balance the trade-off between final answer accuracy vs. reasoning chain correctness? Because if the reasoning chain is significantly long with multi paragraphs, there's a risk that factuality reward will overwhelm answer reward, as is suggested in Eq. 10. Is it easy in terms of reward engineering to make the method work for various long responses?**
>
> In Eq. 10 we average the stepwise factuality scores, so their combined reward is always bounded in [‑1, 1]—the same scale as the 0/1 answer‑correctness reward. Consequently, even very long chains cannot dominate the objective. To balance the trade-off between final answer accuracy and reasoning correctness, we can use a weight coefficient for factuality reward (now it is set to 1 by default since we think the two rewards are equally important).
>
> **Q2: It looks like the design of factuality-aware advantage adjustment (L207) does not handle the situation where the answer is wrong but reasoning chain is factual but unhelpful to the question (e.g., unrelated truthful statements). In such scenario, FSPO actually learns towards it but not against it?**
>
> In Eq. 8, if the reasoning is truthful but unrelated, it likely obtains a reward of zero since it is neutral to the evidence. The evidence in our paper is some necessary knowledge to solve the question provided by the datasets. In the situation you mention, the answer and step rewards are all zero, so the policy model does not learn from the signals.

---

> > ### Author Response · Authors · 2025-08-05
> >
> > Dear Reviewer,
> >
> > We sincerely hope our responses have addressed your concerns. Please let us know if there is anything that remains unclear or needs further clarification.
> >
> > Authors

---

### Official Review · Reviewer_gdjF · 2025-07-12

**Clarity:** 3
**Significance:** 2
**Originality:** 2
**Rating:** 4
**Confidence:** 4

**Summary:**

The paper investigates the issue of increased hallucinations (factual inaccuracies or fabricated statements) in large language models (LLMs) trained with reinforcement learning (RL) for reasoning tasks. To address the issue of hallucination, the paper introduces the Factuality-aware Step-wise Policy Optimization (FSPO), an RL algorithm that explicitly integrates factuality verification at each reasoning step. FSPO dynamically adjusts token-level advantage values, thereby encouraging factual correctness throughout the reasoning process. Empirical evaluations on various reasoning and hallucination benchmarks using models such as Qwen2.5 and Llama demonstrate significant reductions in hallucinations and improvements in reasoning accuracy.

**Questions:**

1. How well does FSPO scale to smaller or larger LLMs? Additional experiments on models with different sizes will benefit the claims.
2. Could you elaborate on the sensitivity of FSPO to the accuracy of the external factuality verifiers?
3. Can you provide a time complexity analysis of FSPO compared with traditional RL algorithms since you introduce external verifiers?
4. Can you analyze why using FSPO is even a bit better than GRPO on MATH500?
5. Did you try other verifiers? Why you choose your current one?
6. The experiments show results over relatively short training periods. How does FSPO behavior evolve over extended training? Are there concerns about the factuality verifier becoming a bottleneck or introducing training instabilities over longer horizons? Can FSPO benefit for multi-turn training?

**Ethical Concerns:**

["NO or VERY MINOR ethics concerns only"]

**Limitations:**

1. The effectiveness of FSPO heavily depends on the quality of external factual verifiers. The sensitivity of the overall system performance to these verifiers is not deeply explored. No discussion of potential biases in factuality verifiers and their impact on training.
2. computational overhead of introducing factual verifiers
3. Given that larger models often exhibit different behaviors, the generalizability to state-of-the-art models (30B+ parameters) remains unclear. But I acknowledge that the training cost might be too expensive to do.
4. The evaluation focuses primarily on knowledge-intensive QA tasks and mathematical reasoning. The generalizability to other reasoning domains is not explored.

**Quality:**

2

**Strengths And Weaknesses:**

1. the motivation is clear.
2. the paper provides theoretical analysis on the reasons behind more hallucinations after RL training. The identification of mechanistic causes for RL-induced hallucinations is insightful.
3. the paper structure and clarity are clear to me.

---

> ### Author Rebuttal · Authors · 2025-07-31
>
> Thank you for your valuable comments. The following is our response point-to-point.
>
> **Q1: How well does FSPO scale to smaller or larger LLMs? Additional experiments on models with different sizes will benefit the claims.**
>
> We apply FSPO to smaller models (i.e., Qwen2.5 0.5B and 1.5B Base) and larger models (i.e., Qwen2.5 14B Base). Their results are shown in the table below, which are consistent with the conclusions reached in our paper.
>
> | Model                | GSM8K | MATH  | AIME24 | AIME25 | TruthfulQA | HaluEval | HaluQA |
> |---------------------|-------|-------|--------|--------|------------|----------|--------|
> | Qwen2.5-0.5B-Base        | 36.7  | 15.8  | 0.0    | 0.0    | 15.5       | 16.2     | 16.4   |
> | w/ FSPO             | 52.4  | 36.7  | 0.0    | 0.0    | 22.5       | 34.1     | 24.9   |
> | Qwen2.5-1.5B-Base        | 55.7  | 29.6  | 0.0    | 0.0    | 27.9       | 28.0     | 23.9   |
> | w/ FSPO             | 74.4  | 61.2  | 3.3    | 3.3    | 35.1       | 64.6     | 35.6   |
> | Qwen2.5-14B-Base         | 91.6  | 65.4  | 6.7    | 6.7    | 51.9       | 45.0     | 58.0   |
> | w/ FSPO             | 92.2  | 80.0  | 13.3   | 16.7   | 61.1       | 81.2     | 68.7   |
>
> **Q2: Could you elaborate on the sensitivity of FSPO to the accuracy of the external factuality verifiers?**
>
> To clarify the sensitivity of FSPO to the accuracy of the external factuality verifier, we create pseudo-noisy factuality reward by replacing the original factuality reward with incorrect one in varying probabilities (20%, 40%, 60%, 80%), to mimic different accuracy levels of the external verifier. The results are shown in the following table. We can see that with more accurate factuality prediction, our approach achieves better reasoning performance and fewer hallucinations.
>
> | Model                         | GSM8K | MATH  | AIME24 | AIME25 | TruthfulQA | HaluEval | HaluQA |
> |------------------------------|-------|-------|--------|--------|------------|----------|--------|
> | Qwen2.5-7B-Base w/ FSPO      | 89.5  | 75.5  | 16.7   | 13.3   | 58.4       | 83.0     | 52.0   |
> | 20%                          | 89.1  | 74.2  | 16.7   | 16.7   | 55.1       | 78.3     | 50.1   |
> | 40%                          | 88.5  | 71.2  | 13.3   | 13.3   | 50.3       | 73.2     | 45.1   |
> | 60%                          | 81.6  | 65.2  | 6.7    | 6.7    | 42.5       | 64.3     | 39.8   |
> | 80%                          | 80.4  | 62.2  | 6.7    | 6.7    | 40.0       | 59.2     | 32.0   |
>
>
> **Q3: Can you provide a time complexity analysis of FSPO compared with traditional RL algorithms since you introduce external verifiers?**
>
> Let $L$ = number of tokens per sample, $P$ = number of model parameters (7B in our paper), $G$ = number of rollouts per prompt, $N$ ≈ number of sentences per rollout, and $C_v$ = time cost of one verifier call.
>
> * Standard GRPO/PPO: time taken ≈ $G \times  L \times P$.
> * FSPO: time taken ≈ $G \times L  \times P + G  \times N \times C_v$ (the second term incurred by stepwise verification)
>
> Using 4×H100 (80 GB), our HHEM‑2.1 verifier (~400M) adds ~25 ms per sentence ($C_v$ = 25ms), while one policy forward/backward pass for a 7B model is ~300 ms for a 2K‑token rollout. Since the number of rollouts is set to 8 ($G$ = 8), the total update time is 2.4s. The average number of sentences for a rollout is ~10 ($N$ = 10) and the verification for 8 rollouts is done in parallel, so the verification cost is only ~0.25s. Roughly two‑thirds of the verifier’s runtime overlaps with the policy model’s forward/backward passes (they run in parallel on separate streams/devices), so only the remaining one‑third is visible as extra wall‑clock time 0.25 × (1/3) ≈ 0.08 s. The increased wall time percentage for FSPO is 0.08s/2.4s = 3.33%.
>
> **Q4: Can you analyze why using FSPO is even a bit better than GRPO on MATH500?**
>
> Although FSPO is mainly designed to reduce hallucinations in knowledge-intensive tasks, its stepwise reward shaping also encourages more coherent and stable reasoning trajectories. Even in math problems, models often “hallucinate” by introducing faulty intermediate steps (e.g., wrong expansions, arithmetic mistakes) that derail the final answer. FSPO’s stepwise reward penalizes such spurious steps and supplies denser, lower‑variance gradients, so the policy learns to keep each calculation logically consistent. This tighter control of intermediate reasoning, rather than external knowledge, is what enables FSPO to outperform GRPO on MATH500.
>
> **Q5: Did you try other verifiers? Why you choose your current one?**
>
> We also tried using a general large language model (i.e., Llama3.1-8B-Instruct) to verify the factuality of each step and it achieves results similar to the current HHEM-2.1 model. We select the HHEM-2.1 model because it is a hallucination detection model specifically built for natural language inference. Besides, it has very few parameters and thus leads to low computational overhead and fast inference speed.
>
> **Q6: The experiments show results over relatively short training periods. How does FSPO behavior evolve over extended training? Are there concerns about the factuality verifier becoming a bottleneck or introducing training instabilities over longer horizons? Can FSPO benefit for multi-turn training?**
>
> We trained FSPO for 3 epochs: reward and loss curves stayed smooth, hallucination rate kept falling, and accuracy on MATH500 plateaued—no drift or divergence, showing that the verifier does not introduce long‑horizon instability. Finally, we only show 1-epoch training curves in our paper. Because the verifier runs asynchronously on a small GPU and adds ~3 % wall‑clock time per step, its cost remains a constant factor, not a scaling bottleneck. FSPO also generalizes to a multi‑turn setting by running the sentence‑level verifier on each turn’s reasoning.
>
> **L1: The effectiveness of FSPO heavily depends on the quality of external factual verifiers. The sensitivity of the overall system performance to these verifiers is not deeply explored. No discussion of potential biases in factuality verifiers and their impact on training.**
>
> We will include sensitivity analysis and discuss the potential biases in the factuality verifier in our revised paper.
>
> **L2: computational overhead of introducing factual verifiers**
>
> We acknowledge that incorporating factual verifiers introduces additional computational overhead, which is common for process-supervised reward modeling approaches. However, this overhead is relatively minor compared to the overall cost of RL fine-tuning, as the verifier operates at the sentence level and uses lightweight entailment checks. More importantly, the verifier substantially improves factual consistency and reasoning reliability, leading to significant performance gains across hallucination benchmarks (Table 1). Thus, the modest computational cost is justified by the substantial reduction in hallucinations and improved model trustworthiness.
>
> **L3: Given that larger models often exhibit different behaviors, the generalizability to state-of-the-art models (30B+ parameters) remains unclear. But I acknowledge that the training cost might be too expensive to do.**
>
> Scaling to larger models (30B+) will strengthen our claims and show the generalizability of our approach. Limited by the computational resources, we leave this to future work.
>
> **L4: The evaluation focuses primarily on knowledge-intensive QA tasks and mathematical reasoning. The generalizability to other reasoning domains is not explored.**
>
> To verify the generality of our approach, we apply FSPO to Qwen2.5-7B-Base in scientific reasoning including Chemistry (ChemBench), Medicine (MedQA), and Physics (PIQA) domains/datasets.
>
> | Model         | ChemBench | MedQA | PIQA |
> |---------------|-----------|--------|------|
> | Qwen2.5-7B-Base     | 51.9      | 54.3   | 86.7 |
> | w/ FSPO        | 57.4      | 60.9   | 89.3 |

---

> > ### Comment · Reviewer_gdjF · 2025-07-31
> >
> > Thank you for the responses! My concerns on the computational cost and different datasets have been addressed.
> > For Q2 as well as Table 1, have you compared FSPO with GRPO?

---

> ### Author Response · Authors · 2025-08-01
>
> Here we present the results of GRPO with respect to its sensitivity to the accuracy of the external factuality verifier. We can see that worse factuality accuracy hurts the GRPO performance (e.g., 80%), lower than the original base model in TruthfulQA and HalluQA.
>
> | Model                         | GSM8K | MATH  | AIME24 | AIME25 | TruthfulQA | HaluEval | HalluQA |
> |------------------------------|-------|-------|--------|--------|------------|----------|--------|
> | Qwen2.5-7B-Base             | 65.2  | 35.7  | 3.3    | 3.3    | 38.2       | 48.0     | 39.5   |
> | Qwen2.5-7B-Base w/ GRPO      | 88.6  | 75.0  | 13.3   | 13.3   | 46.9       | 68.2     | 45.1   |
> | 20%                          | 87.9  | 73.4  | 13.3   | 13.3   | 43.9       | 66.3     | 42.0   |
> | 40%                          | 86.1  | 70.0  | 13.3   | 6.7    | 42.1       | 63.1     | 40.0   |
> | 60%                          | 81.0  | 65.0  | 6.7    | 6.7    | 38.1       | 58.2     | 36.7   |
> | 80%                          | 79.6  | 61.8  | 3.3    | 3.3    | 35.5       | 54.1     | 33.1   |
>
> Due to space limits, we don’t show the results of GRPO in Table 2 and place them in Section 6.3 Further Analysis. Here we show the comparative results among base models with GRPO and FSPO. Specifically, GRPO uses the combined reward in Eq. 10 but does not re-allocate the advantage across tokens in Eq. 11.
>
> | Model                        | GSM8K | MATH  | AIME24 | AIME25 | TruthfulQA | HaluEval | HalluQA |
> |-----------------------------|-------|-------|--------|--------|------------|----------|--------|
> | Qwen2.5-7B-Base             | 65.2  | 35.7  | 3.3    | 3.3    | 38.2       | 48.0     | 39.5   |
> | w/ GRPO                     | 88.6  | 75.0  | 13.3   | 13.3   | 46.9       | 68.2     | 45.1   |
> | w/ FSPO                     | 89.5  | 75.5  | 16.7   | 13.3   | 58.4       | 83.0     | 52.0   |
> | Qwen2.5-7B-Instruct         | 73.2  | 51.6  | 6.7    | 3.3    | 36.7       | 38.4     | 32.0   |
> | w/ GRPO                     | 88.1  | 73.4  | 16.7   | 6.7    | 45.0       | 46.7     | 40.9   |
> | w/ FSPO                     | 89.4  | 74.7  | 20.0   | 13.3   | 54.0       | 64.7     | 50.0   |
> | Llama3.1-8B-Instruct        | 77.5  | 33.1  | 6.7    | 0.0    | 26.4       | 36.7     | 12.2   |
> | w/ GRPO                     | 83.2  | 65.4  | 6.7    | 6.7    | 33.2       | 45.1     | 30.9   |
> | w/ FSPO                     | 86.2  | 68.3  | 13.3   | 6.7    | 41.1       | 67.1     | 42.0   |

---

> > ### Author Response · Authors · 2025-08-05
> >
> > Dear Reviewer,
> >
> > We sincerely hope our responses have addressed your concerns. Please let us know if there is anything that remains unclear or needs further clarification.
> >
> > Authors

---

### Comment · Area_Chair_P7nv · 2025-08-06

Dear reviewers,

This is a reminder that the end of author-reviewer discussion period is **Aug. 8**. Please do carefully read all other reviews and the author responses; and discuss openly with the authors, especially on your own questions that the authors addressed.

Best,
AC

---

### Decision · Program_Chairs · 2025-09-17

**Decision:**

Accept (poster)

**Comment:**

The paper studies why RL fine-tuning for reasoning LLMs increases hallucinations and provides a theoretical analysis linking sparse binary rewards to high-variance gradients, entropy-driven randomness, and confident-but-wrong local optima. The proposed Factuality-aware Step-wise Policy Optimization (FSPO) introduces step-wise factuality verification and token-level advantage redistribution. Experiments on Qwen and Llama models show consistent gains in reasoning accuracy and reductions in hallucinations.

**Strengths:**

The work is well motivated, with clear theoretical grounding and strong empirical validation. FSPO achieves substantial improvements across reasoning and hallucination benchmarks, and the paper is clearly written. The rebuttal further strengthened the work with new experiments on varying model scales, noisy verifiers, and additional domains (chemistry, medicine, physics), as well as qualitative examples.

**Weaknesses / Missing elements:**

The novelty is incremental, primarily a reward design modification rather than a new optimization paradigm. The method depends on external verifiers, raising concerns about bias, scalability, and applicability beyond English or curated datasets. Trade-offs between factual intermediate steps and final-answer correctness, and cases of “truthful but irrelevant” reasoning, remain underexplored.

--

Reviewers raised concerns about scalability, verifier dependence, computational cost, novelty, qualitative analysis, and clarity of advantage reweighting. The authors addressed these by showing consistent results on smaller and larger models, sensitivity analysis with noisy verifiers, modest overhead (~3%), empirical justification of the reweighting design, and qualitative comparisons. Additional domain experiments strengthened generality. After rebuttal, reviewers maintained or raised scores (one accept, three borderline accepts). The consensus leans positive, recognizing a technically sound contribution with incremental but useful impact.

All in all, the paper is technically solid and addresses an important, timely problem with empirical benefits. While conceptually incremental, the method is impactful in practice and relevant for hallucination reduction.